# Role of Autonomous Neuropathy in Diabetic Bone Regeneration

**DOI:** 10.3390/cells11040612

**Published:** 2022-02-10

**Authors:** Johannes Maximilian Wagner, Christoph Wallner, Mustafa Becerikli, Felix Reinkemeier, Maxi von Glinski, Alexander Sogorski, Julika Huber, Stephanie Dittfeld, Kathrin Becker, Marcus Lehnhardt, Mehran Dadras, Björn Behr

**Affiliations:** 1Department of Plastic Surgery, BG University Hospital Bergmannsheil, 44789 Bochum, Germany; max.jay.wagner@gmail.com (J.M.W.); c.wallner88@gmail.com (C.W.); mustafa.becerikli@ruhr-uni-bochum.de (M.B.); felix.reinkemeier@ruhr-uni-bochum.de (F.R.); maxi.vonglinski@bergmannsheil.de (M.v.G.); alexander.sogorski@gmx.de (A.S.); julika.huber@hotmail.com (J.H.); dittfeld.stephanie@gmail.com (S.D.); marcus.lehnhardt@rub.de (M.L.); mdadras@outlook.com (M.D.); 2Department for Orthodontics, University Hospital Düsseldorf, 40225 Dusseldorf, Germany; becker.kathrin@me.com

**Keywords:** diabetic autonomic neuropathy, bone regeneration, fracture healing, diabetes mellitus type 2, sympathetic nervous system, BRL37344

## Abstract

Diabetes mellitus has multiple negative effects on regenerative processes, especially on wound and fracture healing. Despite the well-known negative effects of diabetes on the autonomous nervous system, only little is known about the role in bone regeneration within this context. Subsequently, we investigated diabetic bone regeneration in db^−^/db^−^ mice with a special emphasis on the sympathetic nervous system of the bone in a monocortical tibia defect model. Moreover, the effect of pharmacological sympathectomy via administration of 6-OHDA was evaluated in C57Bl6 wildtype mice. Diabetic animals as well as wildtype mice received a treatment of BRL37344, a β3-adrenergic agonist. Bones of animals were examined via µCT, aniline-blue and Masson–Goldner staining for new bone formation, TRAP staining for bone turnover and immunoflourescence staining against tyrosinhydroxylase and stromal cell-derived factor 1 (SDF-1). Sympathectomized wildtype mice showed a significantly decreased bone regeneration, just comparable to db^−^/db^−^ mice. New bone formation of BRL37344 treated db^−^/db^−^ and sympathectomized wildtype mice was markedly improved in histology and µCT. Immunoflourescence stainings revealed significantly increased SDF-1 due to BRL37344 treatment in diabetic animals and sympathectomized wildtypes. This study depicts the important role of the sympathetic nervous system for bone regenerative processes using the clinical example of diabetes mellitus type 2. In order to improve and gain further insights into diabetic fracture healing, β3-agonist BRL37344 proved to be a potent treatment option, restoring impaired diabetic bone regeneration.

## 1. Introduction

Diabetes mellitus remains a major biomedical burden with an ever-increasing prevalence [1]. As such, in 2017, 430 million people suffered from diabetes mellitus worldwide and this number is expected to increase to 630 million people in 2045 according to the latest Atlas of the International Diabetes Foundation (IDF). [2] As well as affecting multiple organ systems, such as nephropathy or retinopathy, diabetes mellitus is a major cause of impairments in tissue regeneration, such as bone regeneration, wound healing, or development of Charcot foot [3,4,5]. In addition to alterations of angiogenesis in multiple organs, the peripheral nerve system is likewise affected. As such, diabetic polyneuropathy represents the most common neuropathy in the Western World [6]. Diabetic polyneuropathy can be classified into distinct clinical syndromes, whereupon distal symmetrical polyneuropathy is the most common, followed by diabetic autonomic neuropathy (DAN) affecting multiple organs especially by impairment of the sympathetic nervous system (SNS) which often is a subclinical process. Interestingly, SNS fiber loss can be present in up to 50% of patients without clinical evidence of DAN [7]. Alongside the well-known cardiovascular and gastrointestinal neuropathic dysfunctions, DAN facilitates the development of Charcot foot [8]. Bone regeneration in type 2 diabetes is markedly impaired and, besides osteogenesis, it has been shown that impairment of angiogenesis also plays a major role in db^−^/db^−^ (type 2 DM model) mice bone regeneration [9]. The overarching causes remain unanswered; hence, potential explanations include impairment in mobilization of endothelial progenitor cells (EPCs), and subsequently, an impaired function of the bone marrow (BM) stem cell niche [10].

The BM represents a complex environment finely tuned by diverse regulatory mechanisms, which include the SNS. Physiologically, arterioles of the BM are highly innervated by neural fibers of the SNS that regulate the migration of hematopoietic stem and progenitor cells (HSPCs) out of the BM [11], which is regulated by beta3-adrenergic receptors [12]. Furthermore, SNS are known to regulate bone formation by beta2-adrenergic receptors [13]. Confirmed by several in vivo studies, retention of HSPCs in BM niches is primarily dictated by the interaction between the CXCR4 receptor expressed on the surface of the HSPCs and the corresponding ligand CXCL12, namely SDF-1α, released by neighboring supporting (stromal) cells [10,11,14]. Disruption of the CXCL12–CXCR4 axis by reducing expression of both the receptor and ligand enables HSPCs to migrate from the BM to peripheral tissues. Increased SNS activity inhibits CXCL12 synthesis in the neighboring supporting (stromal) cells through activated β3-adrenergic receptors, enabling HSPC egress from the BM.

Lucas and colleagues revealed that cytostatic 5-fluorouracil impairs BM regeneration and HSPC mobilization not only by its direct genotoxic damages induced in the HSPCs, but also by damaging the SNS. Interestingly, protection of SNS fibers through antioxidant treatment with 4-methylcatechol (4-MC) during chemotherapy restored the recovery capacity of the BM completely, emphasizing the significance of SNS input in BM regulation systems and homeostasis [15]. An in vivo study by Albiero et al. finally revealed a profound depletion of SNS fibers induced by hyperglycaemia in the BM of mice with streptozotocin (STZ)-induced diabetes (type 1 DM) and ob^−^/ob^−^ (type 2 DM) mice leading to a remarkable impairment of HSPC mobilization out of the BM [16]. Concordantly, Albiero and colleagues were also able to confirm significantly impaired EPC mobilization in diabetic patients when compared with non-diabetic patients.

There have been additional experimental studies focusing on the relation between SNS activity and mobilization of HSPCs in diabetic BM. However, none of these studies analyzed the involvement of SNS in diabetic bone regeneration [17,18,19]. Moreover, only little is known about the impact of the SNS or the peripheral nervous system, on bone regeneration, irrespective of DM. For instance, in a distraction osteogenesis model, denervation, i.e., complete resection of the sciatic nerve led to impairments of bone regeneration [20]. Few studies indicate that SNS regulates bone formation and resorption during skeletal growth [21] and after trauma in non-diabetic in vivo models [22].

The current understanding of DAN pathophysiology considering that HSPC, including hematopietic stem cells and EPCs, play a crucial role in peripheral tissue repair and regeneration in diabetic conditions [23]. This is an important and novel insight and of great clinical importance which might have an impact on diabetic bone regeneration.

Our aim was to examine the role of autonomic neuropathy for diabetic bone regeneration using an established murine monocortical tibia defect model and, in the second step, to examine the potential therapeutic effect of BRL37344, a β3-adrenergic agonist, in reversing impaired diabetic bone regeneration.

## 2. Materials and Methods

### 2.1. Experimental Animals

All animal experiments were approved by the Institutional AnimalCare and Use Committee, LANUV North-Rhine Westphalia Recklinghausen, Germany. C57BL/6J female littermates at the age of 12 to 14 weeks were used for all experiments. C57BL/6J mice were obtained from Jackson Laboratory (#000664). Heterozygous Leprdb (db^+^/db^−^) mice were obtained from the Jackson Laboratory (catalog no. 000697, Jackson Laboratory, Bar Harbor, ME, USA https://www.jax.org, accessed on 6 June 2021) and kept with access to water and standard laboratory chow ad libitum. Heterozygous db^+^/db^−^ mice on a C57BL/6J background were mated to obtain WT, db^+^/db^−^, and db^−^/db^−^ mice. Genotyping for breeding was performed on genomic DNA by restriction enzyme digest after polymerase chain reaction (forward primer: ATGACCACTACAGATGAACCCAGTCTAC; reverse primer: CATTCAAACCATAGTTTAGGTTTGTC) according to Horvat and Bunger [24].

### 2.2. Pharmacological Treatments

Chemical sympathectomy was performed by intraperitoneal injection of 150 mg/kg 6-hydroxydopamine (6-OHDA; Sigma-Aldrich, St. Louis, MO, USA) in 0.1% ascorbic acid on three consecutive days as described before [25]. Surgical experiments were initiated on the fifth day.

Treatment with the β3-adrenergic agonist BRL37344 (Sigma, St. Louis, MO, USA) was performed by intraperitoneal injections at 2 mg/kg twice per day (every 10–12 h) as reported before [26]. The treatment was started immediately after surgery and continued until euthanasia at day 3 or 7.

### 2.3. Tibial Defect Model

Surgical procedures were performed under inhalation anesthesia with isoflurane and buprenorphine. An established murine tibial defect model was performed as previously described [27]. Briefly, after shaving and disinfecting the left leg, an incision was made on the proximal anterior skin surface over the tibia. After splitting the anterior tibial muscle, the tibia was cautiously exposed. A 1-mm unicortical defect was created on the anterior tibial surface using a drill and saline cooling. The five animal groups included the following: (a) wildtype animals (wt); (b) wildtype animals which have undergone chemical sympathectomy (wt+6OHDA); (c) wildtype animals which have undergone chemical sympthectomy and receive BRL37344 treatment (wt+6OHDA+BRL37344); (d) diabetic animals treated with saline (db); (e) diabetic animals treated with BRL37344 (db+BRL37344). Wound closure was performed with 6-0 Prolene interrupted sutures. The anterior tibial muscle was reset into its anatomical position.

Each group of wildtype and db^−^/db^−^ animals, being histologically analyzed at day 3, consisted of five animals. Each group of animals which were sacrificed 7 days postoperatively for µCT and histological analyses consisted of 7 animals. A total number of 24 db^−^/db^−^ mice and 36 C57BL/6J wildtype mice was used for this work. Euthanasia was performed according to national and international laws and guidelines. Briefly, cervical dislocation was performed after thorough anesthesia to harvest tissue at day 3 and 7 postoperatively. A brief illustration about the different experimental animal groups and time points is given in Figure 1.

### 2.4. Histology

Tibiae of five animals per group were harvested at day 3 and tibiae of 7 animals per group were taken 7 days postoperatively, fixed in 4% paraformaldehyde (Sigma-Aldrich, St. Louis, MO, USA, http://www.sigmaaldrich.com, accessed on 6 June 2021) overnight at 4 °C, and decalcified in 19% EDTA (PanReac Applichem, Darmstadt, Germany, https://www.applichem.com, accessed on 6 June 2021) for 5 days with daily changes of solution. Samples were then dehydrated, embedded in paraffin, and cut into serial sagittal sections (thickness 6–9 µm). Aniline-blue and TRAP stainings were performed according to the manufacturer’s instructions. For immunohistochemical staining, sections were incubated at 58 °C for 1 h and subsequently rehydrated and incubated with 0.125% Proteinase K for 30 min. After a short washing step with PBS, sections were permeabilized with 0.1% Tween 20 for 4 min and treated with blocking solution for 1 h. Incubation with primary antibodies against Tyrosinehydroxilase (rabbit, polyclonal; Santa Cruz Biotechnology, Santa Cruz, CA, USA, http://www.scbt.com, accessed on 6 June 2021; sc-10758, 1:50, RUNX-2, AB_2184247), SDF-1 (Santa Cruz Biotechnology, Santa Cruz, CA, USA; sc-28876, 1:50) or Runx2 (Santa Cruz Biotechnology, Santa Cruz, CA, USA; sc-10758, 1:50) followed overnight in blocking solution at 4 °C. After washing with PBS, secondary antibody (goat anti-rat conjugated with Alexa 594, Thermo Fisher Scientific Life Sciences, Waltham, MA, USA 1:1000 dilution in PBS) has been applied and incubated for 4 h at room temperature. All sections were counterstained with 49,6-diamidino-2-phenylindole (DAPI). Sections were subsequently mounted with Fluoromount Aqueous Mounting Medium (Sigma-Aldrich). Images were taken with a fluorescence microscope (model IX83, Olympus, Tokyo, Japan, http://www.olympus.co.jp, accessed on 6 June 2021).

### 2.5. Image Quantification

Histological images were quantified via selection of a 2000 × 2000 Px region and semiautomatic selection of positive stained pixels via magic wand tool (tolerance 60%; noncontiguous) using Adobe Photoshop (Adobe, San José, CA, USA). For each staining, 4 slides were selected and quantified per sample.

### 2.6. Microcomputed Tomographic Analysis (µCT)

Tibiae of 7 animals per group were taken 7 days after surgery and scanned with a μCT device (Viva CT 80; Scanco Medical AG, Brüttisellen, Switzerland). The scan was operated at 70 kVp, 114 μA, 8 W, 31.9-mm FOV, an integration time of 1167 ms and 2× frame averaging. Reconstruction of data sets was performed into 3D volumes with an isotropic nominal resolution of 15.6-μm voxel size.

### 2.7. Image Processing

Image processing was performed by using μ-CT Evaluation Software Program V6.5 (Scanco Medical AG, Brüttisellen, Switzerland). For evaluation, a standardized cylindrical volume of interest with 16.84 mm in diameter was placed within the region of interest of the defect site. An assessment of bone volume to total volume (BV/TV) was performed according to the guidelines for assessment of bone microstructures using μCT [28].

### 2.8. Statistical Methods

Presented results were shown as mean ± standard deviation (SD). Comparing two groups, *p*-values were calculated using Student’s *t*-test and ANOVA was used for comparison of more than two groups. Statistical significance was set at a *p*-value < 0.05.

## 3. Results

### 3.1. Activity of Sympathetic Nervous System in Bone of db^−^/db^−^ Mice Is Decreased

Immunofluorescence of bone marrow for tyrosinehydroxlase as an indicator of sympathetic activity revealed a marked decrease in db^−^/db^−^ mice compared with wildtype controls (Figure 2). Additionally, day 7 analysis of wildtype mice after chemical sympathectomy with 6-OHDA confirmed successful silencing of the sympathetic nervous system with a significant decrease in tyrosinehydroxylase in immunofluorescence staining. As expected, treatment with BRL37344 did not significantly change the quantity of tyrosinehydroxylase fibers in sympathectomized and diabetic mice (Figure 2). Furthermore, the downstream target of sympathetic nervous system SDF-1 was significantly downregulated in diabetic and sympathectomized animals and successfully upregulated by BRL37344 treatment. Interestingly, SDF-1 signaling in BRL37344 treated animals showed similar levels as untreated wildtype control.

### 3.2. Chemical Sympathectomy Reduces Bone Regeneration in Wildtype Mice Comparable to db^−^/db^−^ Mice

Chemical sympathectomy via administration of 6-OHDA lead to a significant decrease in bone formation at day 7 in WT mice compared with PBS controls. This could be shown by Aniline-blue, Masson–Goldner histology stains (Figure 3). This decrease in bone formation was comparable to significantly reduced bone regeneration in db^−^/db^−^ mice (Figure 3). Furthermore, µCT scans further validated these observations, showing a significantly decreased bone volume to tissue volume ratio in defect areas of sympathectomized and diabetic animals (Figure 4).

### 3.3. BRL37344 Treatment Rescues Bone Regeneration of db^−^/db^−^ Mice and Sympathectomized Wildtype Mice

Treatment of db^−^/db^−^ mice with the β3-adrenergic agonist BRL37344 leads to a significant increase in bone regeneration, demonstrated in Aniline-Blue and Masson–Goldner staining, almost reaching WT levels (Figure 3). Accordingly, BRL37344 treatment was able to restore bone regeneration in sympathectomized WT mice at day 7 in Aniline-Blue and Masson–Goldner staining (Figure 3). Accordingly, µCT analyses proved a successful treatment of sympathectomized wildtype and diabetic animals via BRL37344, significantly increasing bone volume to tissue volume ratio (Figure 4).

### 3.4. Diminished Osteoblastogenesis and Bone Turnover Is Being Restored in Diabetic and Sympathectomized Animals by BRL37344 Treatment

Having shown reduced bone regeneration in diabetic and sympathectomized wildtype animals, further analysis focused on bone homeostasis of osteoclasts and osteoblasts. TRAP staining revealed decreased osteoclasts and thus decreased bone turnover in diabetic and sympathectomized animals at day 7 postoperatively (Figure 5). Furthermore, reduced Runx2 signaling in immunofluorescence at day 3, as an important marker for osteoblastogenesis, was observed in both groups (Figure 5). Diminished bone turnover, as well as reduced osteoblastogenesis could be fully restored comparable to levels of uninjured wildtypes in both BRL37344 treatment groups. Interestingly, Runx2 signaling was nearly absent in 6-OHDA wildtype animals.

## 4. Discussion

We were able to show that diabetic autonomic neuropathy directly affects bone regeneration by harming the autonomic nervous system of the bone. Thus, this mechanism might be one potential reason for decreased bone regeneration upon diabetic conditions. Furthermore, chemical sympathectomy leads to decreased bone regeneration in a murine unicortical tibia defect model. As a downstream mediator of beta3-agonism, SDF-1 was decreased after sympathectomy or under diabetic conditions and increased by BRL37344-treatment. Bone regeneration and bone turnover of diabetic animals as well as sympathectomized animals could be restored by application of beta3-agonist BRL37344.

The sympathetic nervous system has various regulatory functions in regenerative processes and its disruption can lead to a delay or even disable these processes. For example, Kim et al. have reported that chemical sympathectomy via 6-OHDA leads to delayed wound healing in rats [29]. Saburo et al. showed, that chemical sympathectomy might lead to a decreased collagen metabolism and decreased capillaries in an animal burn model [30].

Furthermore, it has various roles in bone metabolism. Bataille et al. have already shown that bone of the appendicular skeleton was mostly catecholaminergic dependent, while mandibular bone was more cholinergic dependent [31]. According to our results, they had shown that sympathectomy reduces number of TRAP-positive cells as a sign of reduced bone metabolism and turnover. An important mechanism for beta3-receptor induced effects on bone metabolism has been discovered by Vafaeil et al., who showed a modulation of the MSC niche, resulting in increased SDF-1 expression and improved MSC colony forming potency, via the administration of BRL37344 in a murine animal model [32].

In previous studies, we had shown that diabetic conditions lead to worse bone regeneration and angiogenesis in the established unicortical tibia defect model [9,33]. According to the present results, this was mainly a cause of reduced osteoblastogenesis. Accordingly, Enriquez et al. induced type 1 diabetes in rodents by streptozocin injection and could proof a decreased bone trabecular thickness associated with decreased tyrosinhydroxylase positive sympathetic nerve fibers [34]. Very recently, Bubb et al. were able to show that beta 3-agonist treatment markedly increases angiogenesis in a murine diabetic model of peripheral arterial disease [35]. Interestingly, osteoblast like UMR106 cells cultivated with beta 3-agonist BRL37344 showed downregulation of Runx2 and Dlx5 expression but increased osteocalcin and osteopontin [36], which is partly in contrast to our in vivo data but undermines the ambiguous regulation of osteoblasts in vitro and in vivo.

There are serious limitations of this study which have to be addressed. The presented data indicate an impact of the autonomic nervous system on diabetic bone regeneration, but there is limited validity about the exact mechanisms underlying these observations, as these primary data are predominantly descriptive. Further studies are needed to confirm these data and gain further insights into the pathomechanisms of diabetic fracture healing and autonomic nervous regulation of bone healing.

Although not fully proven, there are several human cohort analyses which strongly indicate a correlation between diabetic polyneuropathy and thereby disruption of the sympathetic nervous system and the impairment of bone metabolism. Quite recently, a cohort analysis of overweight, middle-aged individuals in Australia did show a correlation between a decreased bone mass and a reduced sympathetic nervous system activity [37]. Comparing human bone specimens of ankle joint arthritis and diabetic Charcot foot disease, Koeck et al. could recognize minor signaling of sympathetic nerve fibers with the latter [38].

## 5. Conclusions

The burden of diabetic patients suffering from fractures is a severe clinical problem. In the present study, the importance of the sympathetic nervous system for bone regeneration, and especially for diabetic fracture healing, has been demonstrated. Moreover, a novel and innovative therapeutic option is being presented, which restores impaired osteogenesis of diabetic fracture healing, via beta3-receptor agonism stimulating the autonomous nervous system. For diabetic patients these results are of great importance not only considering impaired acute fracture healing, but moreover, the prevention of bone fractures, especially in patients with long lasting diabetes. Based on this work, further studies should be performed, validating these experimental data in humans and developing an effective non-surgical treatment for diabetic bone regeneration as a future perspective.

## Figures and Tables

**Figure 1 cells-11-00612-f001:**
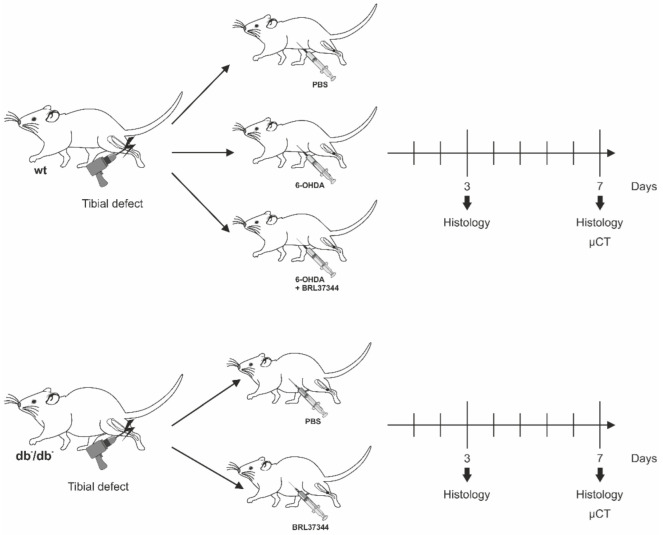
Illustration of experimental groups: wt: wildtype mouse; db^−^/db^−^: diabetic mouse; wildtype mice were sympathectomized via application of 6-OHDA. Both wt and db^−^/db^−^ mice received BRL37344 treatment and were analyzed at days 3 and 7. Control animals were treated with PBS.

**Figure 2 cells-11-00612-f002:**
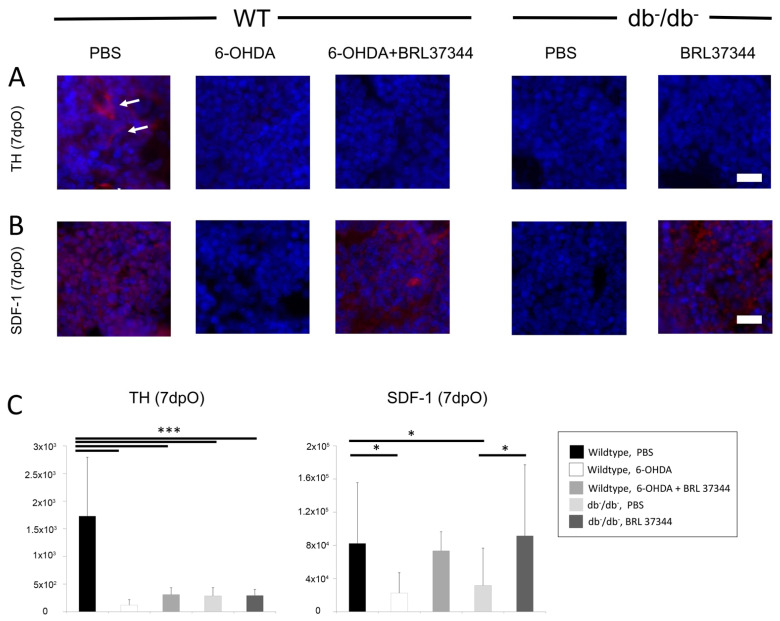
Immunohistochemistry for (**A**) tyrosinhydroxylase (TH) and (**B**) SDF-1, 7 days postoperatively. Histomorphometrical analysis (**C**) of TH and SDF-1. Staining of TH depicts significantly decreased signaling of all sympathectomized wildtype (6-OHDA) and diabetic groups, while SDF-1 shows reduced levels in 6-OHDA wildtype and db^−^/db^−^ animals, which could be fully restored by BRL37344 treatment. White arrows indicate signaling of TH staining. Y-Axis shows pixel counts. Results are shown as mean ± SD. *p*-value: * < 0.05, *** < 0.001. Scale bar: 200 μm.

**Figure 3 cells-11-00612-f003:**
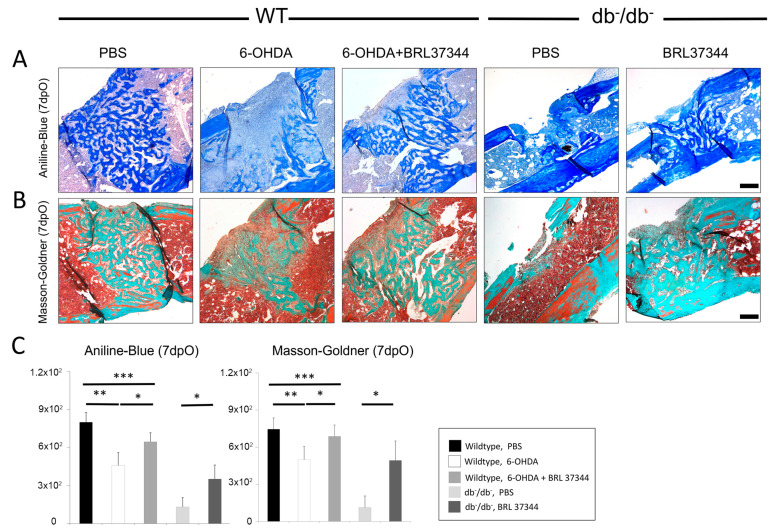
(**A**) Aniline-Blue and (**B**) Masson–Goldner staining with (**C**) histomorphometrical analyses, 7 days postoperatively. Aniline-Blue and Masson–Goldner staining both indicated significantly reduced bone regenerative capacity in sympathectomized wildtype (wt, 6-OHDA) and diabetic mice (db^−^/db^−^). In both experimental groups BRL37344 treatment proved to be successful in restoring bone regeneration. Y-axis shows pixel counts in Aniline-Blue and Masson–Goldner staining. Results are shown as mean ± SD. *p*-value: * < 0.05, ** < 0.01, *** < 0.001. Scale bar: 200 μm.

**Figure 4 cells-11-00612-f004:**
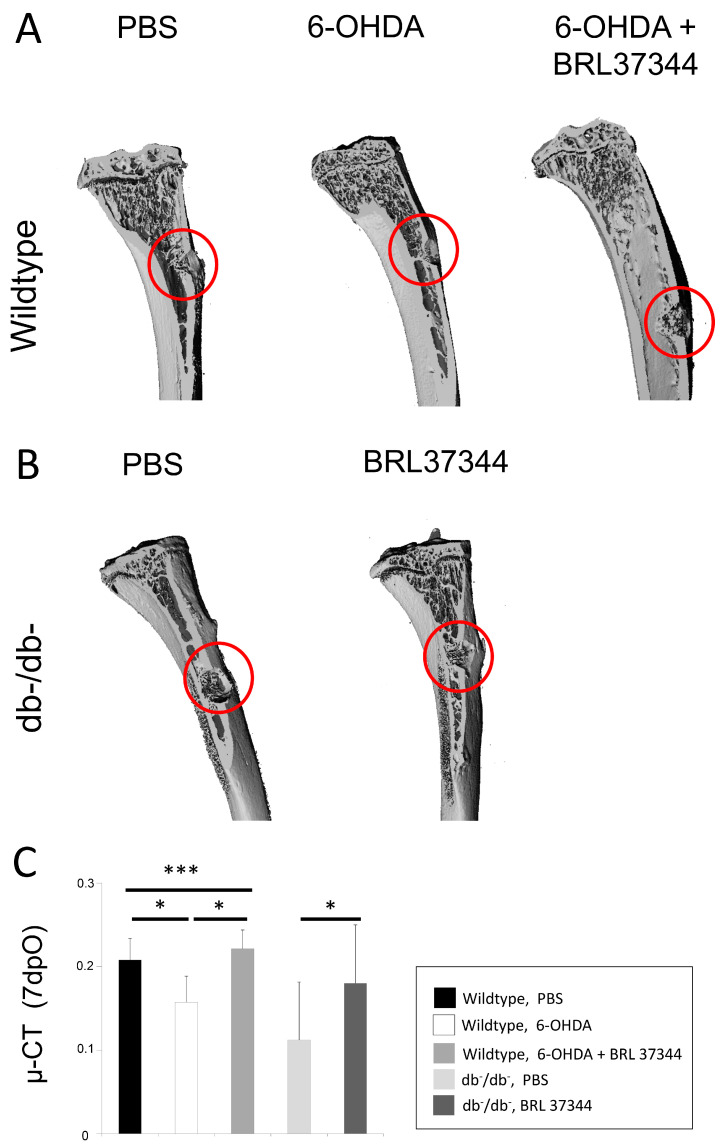
µCT-scans, 7 days postoperatively of (**A**) wildtype and (**B**) db^−^/db^−^ mice. µCT further validate results of histological analyses, revealing decreased bone volume to tissue volume ratio (**C**) in 6-OHDA wildtype and db^−^/db^−^ animals, being significantly enhanced by BRL37344 treatment. Y-axis shows bone volume to total volume (bv/tv). Results are shown as mean ± SD. *p*-value: * < 0.05, *** < 0.001.

**Figure 5 cells-11-00612-f005:**
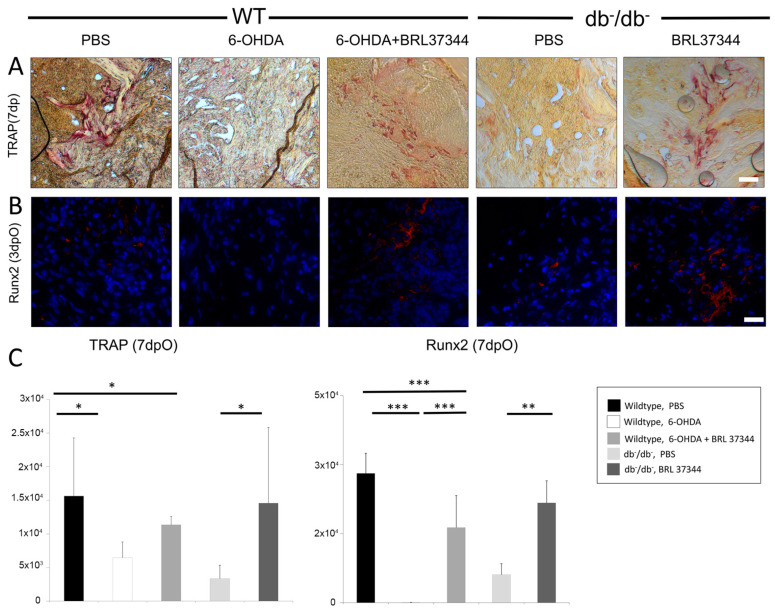
(**A**) TRAP staining (7 days postoperatively) and (**B**) immunohistochemistry for Runx2 (3days postoperatively) with (**C**) histomorphometrical analyses. Reduced bone turnover is seen in TRAP staining due to chemical sympathectomy in wildtype mice and in diabetic animals. Runx2 staining indicate significantly impaired osteoblastogenesis in 6-OHDA wildtype and db^−^/db^−^ groups. Both BRL37344 treatment groups showed significant enhancement of osteoblastogenesis and bone turnover, almost reaching physiological levels. Y-axis shows pixel counts. Results are shown as mean ± SD. *p*-value: * < 0.05, ** <0.01, *** < 0.001. Scale bar: 50 μm.

## Data Availability

The data that supports the findings of this study are available in the figures of this article.

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
