# Peer review of "Role of Autonomous Neuropathy in Diabetic Bone Regeneration"

_cells, 2022, doi:10.3390/cells11040612_

Round 1

Reviewer 1 Report

The article entitled “Role of autonomous neuropathy in diabetic bone regeneration” , this study aimed to was to examine the role of autonomic neuropathy for diabetic bone   regeneration using an established murine monocortical tibia defect model and in the second

step examine the potential therapeutic effect of BRL37344, a β3-adrenergic agonist, in re-

versing impaired diabetic bone regeneration.

Below are some suggestions:

In the Abstract:

- The abstract is well written, but the conclusion can be more enlightening and affirmative in relation to the research objective.

- I suggest better describing the methodology

In the Introduction:

- In general, the introduction is long and a little confusing, I suggest organizing the ideas better as well as emphasizing the clinical importance of the research carried out.

In the Materials and Methods:

- The methodology was well described

- Could the authors clarify why the morphometric analysis of the histological slides was not performed?

- 2.1 Experimental animals: I suggest a better description of the groups....how many animals were used? An illustrative image of the randomization of the groups could be made as well as the experimental design.

In the Results:

- 3.3. BRL37344 treatment rescues bone regeneration of db-/db- mice and sympathectomized wildtype mice: In figure 2 I suggest making two plates separating the histological images from the microtomography images for better visibility and quality.

In the Discussion:

- The discussion could be written according to and in the same order as the data presented in the result, including the limitations of the study.

In the Conclusion:

- Change the conclusion into final considerations, including the purpose of the research, its main results, the conclusions and future clinical perspectives.

Author Response

We would like to thank the reviewer and appreciate the invested time and comments for our manuscript. We will now respond to the reviewers comments point-by-point:

Reviewer 1:

The article entitled “Role of autonomous neuropathy in diabetic bone regeneration” , this study aimed to was to examine the role of autonomic neuropathy for diabetic bone   regeneration using an established murine monocortical tibia defect model and in the secondstep examine the potential therapeutic effect of BRL37344, a β3-adrenergic agonist, in reversing impaired diabetic bone regeneration.

Below are some suggestions:

 In the Abstract:

- The abstract is well written, but the conclusion can be more enlightening and affirmative in relation to the research objective.

- I suggest better describing the methodology

 Our answer:

Thank you for your comments. The Abstract has been revised according to your suggestions in the revised manuscript.

In the Introduction:

- In general, the introduction is long and a little confusing, I suggest organizing the ideas better as well as emphasizing the clinical importance of the research carried out.

Our answer:

We revised and reorganized the introduction part of the revised manuscript and hopefully made the ideas and the clinical aspect of our work more visible and less confusing. 

In the Materials and Methods:

- The methodology was well described

- Could the authors clarify why the morphometric analysis of the histological slides was not performed?

Our answer:

We apologize for the missing information in the methods section. We performed morphometric analysis of all histological slides, using semiautomated pixel quantification. We added relevant information in the methods section of the revised manuscript. 

- 2.1 Experimental animals: I suggest a better description of the groups....how many animals were used? An illustrative image of the randomization of the groups could be made as well as the experimental design.

 Our answer:

We thank the reviewer for this suggestion. We used five animals per group which were euthanized 3 days postoperatively and 7 animals per group 7 days postoperatively. We included missing information in the methods section and furthermore included an illustration about the experimental design, which we named figure 1 in the revised manuscript.

In the Results:

- 3.3. BRL37344 treatment rescues bone regeneration of db-/db- mice and sympathectomized wildtype mice: In figure 2 I suggest making two plates separating the histological images from the microtomography images for better visibility and quality.

 Our answer:

We made two separate images dividing histological images and µCT, which we named figure 3 and 4 in the revised manuscript. 

In the Discussion:

- The discussion could be written according to and in the same order as the data presented in the result, including the limitations of the study.

 Our answer:

The discussion part of the revised manuscript has been restructured according to the results presented. Furthermore, limitations of the presented work have been added.  

In the Conclusion:

- Change the conclusion into final considerations, including the purpose of the research, its main results, the conclusions and future clinical perspectives.

Our answer:

The conclusion has been revised according to your suggestions.

Reviewer 2 Report

The topic of this paper is clinically very important and its significance is properly described in the Introduction. The approach used by authors is very interesting and the conducted experiments have elucidated potential mechanism of disrupted bone regeneration in diabetic patients.

However, description of conducted methods and presentation of results should be significantly improved and several issues should be addressed:

  1. The experimental design is not properly described. Please indicate how many animals were terminated in each time point and which analyses were conducted in each time point. I would suggest to add a diagram/schematic presentation of the study experimental design. 
  2. What was the number of animals in each experimental group? Please clarify the sentence "Each group consisted of at least five animals"; was the number of animals different between the groups? Was the same number of animals terminated at day 3 and day 7? What was the total number of animals used in the study? Please include these information in the proper section in M&M
  3. How was histomorphometrical analysis / quantification of histology conducted? Although it is the main method used in the research it is completely ommited in the M&M section? Please describe it in detail and include information regarding the number of samples and sections per each sample
  4. The presentation of the results is the weakest point of the paper and should be significantly improved. Textual description of the results is too short (4.5 lines of text per section) and should be expanded. 
  5. Figure 1: Please add captions (A, B) to different parts of the figure and describe in the legend what is shown in each part of the figure. In the first part of the figure it is very diffucult to see differences among experimental groups. I would suggest to improve the picture quality/contrast or add arrows. As mentioned before it is not clear how numerical values were calculated. 
  6. Figure 2: Please add captions (A, B, C and D) to different parts of the figure and describe in the legend what is shown in each part of the figure. Same as on previous figure it is not clear how numerical values (for histological sections) were calculated. 
  7. Figure 3: Please add captions (A, B and C) to different parts of the figure and describe in the legend what is shown in each part of the figure. Same as on previous figure it is not clear how numerical values (for histological sections) were calculated. 

Minor issues:

Line 138: "... cut into serial saggital sections (thickness 6-9 mm)." I believe that the thickness of sections is probably 6-9 micrometers instead of 6-9 mm.

I would suggest to choose either "MicroCT" or "μCT" spelling and use it consistently through the text.

Author Response

We would like to thank the reviewer and appreciate the invested time and comments for our manuscript. We will now respond to the reviewers comments point-by-point:

Reviewer 2

The topic of this paper is clinically very important and its significance is properly described in the Introduction. The approach used by authors is very interesting and the conducted experiments have elucidated potential mechanism of disrupted bone regeneration in diabetic patients.

However, description of conducted methods and presentation of results should be significantly improved and several issues should be addressed:

The experimental design is not properly described. Please indicate how many animals were terminated in each time point and which analyses were conducted in each time point. I would suggest to add a diagram/schematic presentation of the study experimental design.

Our answer:

We added relevant information in the methods section and furthermore, included an illustration about the experimental design, which we named figure 1 in the revised manuscript.

  1. What was the number of animals in each experimental group? Please clarify the sentence "Each group consisted of at least five animals"; was the number of animals different between the groups? Was the same number of animals terminated at day 3 and day 7? What was the total number of animals used in the study? Please include these informations in the proper section in M&M

Our answer:

Dear reviewer, we apologize for the insufficient information provided in the methods section. Each group consisted of five animals at day 3 and 7 animals at day 7. The total number of animals used were 36 wildtype and 24 db-/db- mice, as one bone specimen could be used for µCT as well as histological analyses. We added missing information in the methods section of the revised manuscript.

  1. How was histomorphometrical analysis / quantification of histology conducted? Although it is the main method used in the research it is completely ommited in the M&M section? Please describe it in detail and include information regarding the number of samples and sections per each sample

Our answer:

We apologize for the missing information and added relevant information in the revised M&M section under the subheading "2.5 Image quantification".

  1. The presentation of the results is the weakest point of the paper and should be significantly improved. Textual description of the results is too short (4.5 lines of text per section) and should be expanded. 

Our answer:

We apologize for the insufficient presentation of the results and overhauled the whole section in the revised manuscript.

  1. Figure 1: Please add captions (A, B) to different parts of the figure and describe in the legend what is shown in each part of the figure. In the first part of the figure it is very diffucult to see differences among experimental groups. I would suggest to improve the picture quality/contrast or add arrows. As mentioned before it is not clear how numerical values were calculated. 

Our answer:

We revised figure 2 in the revised manuscript and added captions and arrows according to your suggestions. Furthermore, we added missing information in the methods section under the subheading "image quantification".

  1. Figure 2: Please add captions (A, B, C and D) to different parts of the figure and describe in the legend what is shown in each part of the figure. Same as on previous figure it is not clear how numerical values (for histological sections) were calculated. 

Our answer:

Your suggestions have been implemented in revised figure 3 and 4 of the revised manuscript. 

  1. Figure 3: Please add captions (A, B and C) to different parts of the figure and describe in the legend what is shown in each part of the figure. Same as on previous figure it is not clear how numerical values (for histological sections) were calculated. 

Our answer:

According to your comment figure 5 in the revised manuscript has been changed.

Minor issues:

Line 138: "... cut into serial saggital sections (thickness 6-9 mm)." I believe that the thickness of sections is probably 6-9 micrometers instead of 6-9 mm.

Our answer:

We apologize for the mistake and corrected the unit accordingly.

I would suggest to choose either "MicroCT" or "μCT" spelling and use it consistently through the text.

Our answer:

We revised the manuscript and changed the spelling to µCT.

Round 2

Reviewer 1 Report

All suggestions were made in the article.

Author Response

We thank the reviewer for the useful comments, improving our revised manuscript.

Reviewer 2 Report

The authors have made all suggested corrections and significantly improved the manuscript which is now suitable for publication. 

Author Response

(The authors gave the same response as above.)
